# Systematic Review of the Potential of MicroRNAs in Diffuse Large B Cell Lymphoma

**DOI:** 10.3390/cancers11020144

**Published:** 2019-01-26

**Authors:** Ane Larrabeiti-Etxebarria, Maria Lopez-Santillan, Borja Santos-Zorrozua, Elixabet Lopez-Lopez, Africa Garcia-Orad

**Affiliations:** 1Department of Genetics, Physical Anthropology and Animal Physiology, Faculty of Medicine and Nursing, University of the Basque Country (UPV/EHU), 48940 Leioa, Spain; anelarrabeiti@gmail.com (A.L.-E.); lopezsantillanmaria2@gmail.com (M.L.-S.); Borja.santos@ehu.eus (B.S.-Z.); africa.garciaorad@ehu.eus (A.G.-O.); 2Pharmacy Service, Araba University Hospital-Txagorritxu, 01009 Vitoria, Spain; 3Medical Oncology Service, Basurto University Hospital, 48013 Bilbao, Spain; 4BioCruces Health Research Institute, 48903 Barakaldo, Spain

**Keywords:** lymphoma, microRNA, diagnosis, classification, treatment response, prognosis

## Abstract

Diffuse large B cell lymphoma (DLBCL) is the most common subtype of invasive non-Hodgkin’s lymphoma (NHL). DLBCL presents with variable backgrounds, which results in heterogeneous outcomes among patients. Although new tools have been developed for the classification and management of patients, 40% of them still have primary refractory disease or relapse. In addition, multiple factors regarding the pathogenesis of this disease remain unclear and identification of novel biomarkers is needed. In this context, recent investigations point to microRNAs as useful biomarkers in cancer. The aim of this systematic review was to provide new insight into the role of miRNAs in the diagnosis, classification, treatment response and prognosis of DLBCL patients. We used the following terms in PubMed” ((‘Non-coding RNA’) OR (‘microRNA’ OR ‘miRNA’ OR ‘miR’) OR (‘exosome’) OR (‘extracellular vesicle’) OR (‘secretome’)) AND (‘Diffuse large B cell lymphoma’ OR ‘DLBCL’)” to search for studies evaluating miRNAs as a diagnosis, subtype, treatment response or prognosis biomarkers in primary DLBCL in human patient populations. As a result, the analysis was restricted to the role of miRNAs in tumor tissue and we did not consider circulating miRNAs. A total of thirty-six studies met the inclusion criteria. Among them, twenty-one were classified in the diagnosis category, twenty in classification, five in treatment response and nineteen in prognosis. In this review, we have identified miR-155-5p and miR-21-5p as miRNAs of potential utility for diagnosis, while miR-155-5p and miR-221-3p could be useful for classification. Further studies are needed to exploit the potential of this field.

## 1. Introduction

Diffuse large B-cell lymphoma (DLBCL) is the most common lymphoid malignancy in adults, accounting for 30–40% of all non-Hodgkin lymphoma cases [1]. DLBCL presents a very diverse clinical and genetic background, which leads to highly heterogeneous outcomes among patients [2]. 

The first line therapy for this aggressive malignancy consists of combined chemotherapy including rituximab, prednisone, doxorubicin, vincristine and cyclophosphamide (R-CHOP). Around 75–80% of patients achieve complete remission with R-CHOP therapy. Nevertheless, approximately up to 40 % of patients have primary refractory disease or relapse [3]. Worryingly, those patients tend to respond poorly to additional chemotherapy lines, which remains a major cause of morbidity and mortality [4].

In order to identify DLBCL patients with a higher risk of a poor response to therapy, different tools have been developed. The International Prognostic Index (IPI) predicts overall and progression free survival based on five risk factors: age, tumor stage, serum lactate dehydrogenase (LDH) concentration, performance status and number of extra nodal disease sites [5]. Using those factors, the IPI distinguishes four risk groups with different 5-year overall survival, ranging from 26 to 73% [6]. Nevertheless, some patients present an unfavorable course of disease despite having a good prognostic index. Another profiling tool uses gene expression profiling (GEP) or immunohistochemical analysis to define two molecular subtypes with different clinical outcomes independent of IPI stratification: the germinal center B-cell-like (GCB) DLBCL, and the activated B-cell-like (ABC) DLBCL. The 5-year survival rates are 60% for GCB and 40% for ABC subtype [7,8]. However, it is not possible to identify all the patients that will not respond to therapy with these profiling tools. Therefore, new biomarkers are needed for a better patient stratification. 

In this sense, important knowledge is emerging regarding novel molecular and biological candidates with diagnostic, predictive and prognostic potential in DLBCL, including microRNAs (miRNAs) [9]. MiRNAs are small, non-coding RNAs with a role in gene expression regulation at the post-transcriptional level. They bind the 3′ untranslated region (UTR) of a target mRNA, which leads to their repression or degradation [10]. Through this mechanism, miRNAs regulate more than 50% of known human genes [11], including genes of the 10 main routes involved in cancer [12]. 

Accordingly, recent research has shown the potential role of miRNAs as diagnostic, classification and prognostic predictors in cancer [13]. For instance, miR-21 has been intensively studied as a diagnosis tool, being found upregulated in many types of cancer including non-small cell lung cancer (NSCLC) [14]. Other examples include miR-10b, which has been described as a useful tool for the classification of papillary renal cell carcinoma type 2 [15], and miR-183, high expression levels of which have been associated with poor prognosis in different cancer types such as colorectal cancer, pancreatic cancer, lung cancer, gastric cancer, and breast cancer [16]. 

Abnormal expression of miRNAs is also common in B cell neoplasms, including B cell lymphoma. However, there is inconsistency in the data reported. Consequently, the aim of this systematic review was to clarify the role of deregulated miRNAs in DLBCL tumor samples as more systematically-defined diagnostic, subtype, prediction of treatment response and prognostic biomarkers.

## 2. Results

The detailed search results are included in Figure 1. In brief, the search strategy provided a total of 508 records in the PubMed database. Once the duplicated articles were removed, 338 remained. Of these 338, 239 were excluded after reading the abstract because they did not meet the inclusion criteria. Then, the full texts of the remaining 99 studies, which focused on miRNAs in DLBCL, were read carefully. Additionally, another 63 articles were excluded because there were other coexisting pathologies, miRNAs were not analyzed in the tumor sample, DLBCL was not primarily considered, they did not assess the role of miRNAs in diagnosis, subtype, prediction of treatment response or prognosis, or miRNA expression changes were not considered. A total of 36 studies investigating the role of miRNA expression changes as biomarkers in DLBCL tumor samples were included. Twenty-one of them considered miRNAs as putative DLBCL diagnosis biomarkers, twenty in subtype classification, five in treatment response and nineteen of the studies searched for markers for their role in prognosis.

### 2.1. Tumor Tissue miRNAs as Biomarkers for Diagnosis in DLBCL

Twenty-one studies analyzed the expression of miRNAs by comparing DLBCL cases vs healthy controls [17,18,19,20,21,22,23,24,25,26,27,28,29,30,31,32,33,34,35,36,37]. These 21 studies provided a total of 140 differentially expressed microRNAs in DLBCL patients compared with healthy control individuals as shown in Appendix A.

Regarding the miRNAs that were concordantly deregulated in more than two studies, we identified two miRNAs that were repeatedly reported to be up-regulated in DLBCL patients (miR-155-5p [17,22,24,26,28,30,32,33,34], miR-21-5p [19,22,26,28,33,36], although some studies did not find a significant association (miR-155-5p [27,29], miR-21-5p [27,32]).We also identified two miRNAs with contradictory results. On the one hand, miR-150-5p was found to be down-regulated in DLBCL patients in four studies [18,28,31,32] and contradictorily up-regulated in DLBCL patients in another study [29], while no significant association was reported in the remaining study [27]. On the other hand, miR-146a/b-5p was found upregulated in three studies [22,27,30] while it was shown to be discordantly downregulated in another study [21], and not significantly associated with DLBCL in two studies [27,28] (Table 1).

### 2.2. Tumor Tissue miRNAs as Biomarkers for DLBCL Subtype Classification

Twenty studies analyzed the role of tumor tissue miRNAs to distinguish between GCB and ABC DLBCL subtypes and their characteristics are provided in Appendix A [19,21,23,24,25,26,27,28,30,32,33,34,38,39,40,41,42,43,44,45]. These studies found 93 miRNAs differentially expressed between GCB and ABC DLBCL samples. Among these 93 differentially expressed miRNAs, five miRNAs were concordantly reported in more than two studies. Four of them were reported as down-regulated (miR-155-5p [26,27,28,30,33,34,38,41,44], miR-221-3p [27,33,41,45], miR-222-3p [27,41,45], and miR-146a/b-5p [28,30,41]) or unchanged (miR-221-3p [28,43], miR-222-3p [28,32,38,43] and miR-146a/b-5p [21,27,28,38,43]) in GCB samples, whereas miR-28-5p was found to be up-regulated [28,41] or unchanged [28,41] in the same subtype (Table 2).

### 2.3. Tumor Tissue miRNAs as Biomarkers for Prediction of Treatment Response in DLBCL

Five studies were focused in the role of miRNAs in DLBCL tissue as predictive biomarkers of response to R-CHOP treatment [18,23,24,30,42]. The characteristics of each study are shown in Appendix A. A total of five miRNAs were differentially expressed between good and poor responders. Three microRNAs were found to be associated with a favorable response to therapy (miR-27-3p [18], miR-34a-5p [24] and miR-224-5p [23]), whereas miR-155-5p [30] and miR-146-5p [30] were found to be associated with chemoresistance. However, each miRNA was analyzed in only one study, without any of the results being replicated.

### 2.4. Tumor Tissue miRNAs as Biomarkers for DLBCL Prognosis

The relevance of tumor tissue miRNAs for prognosis in DLBCL patients was analyzed in nineteen studies [18,20,21,23,25,26,27,30,32,33,37,38,41,42,45,46,47,48,49]. A total of 50 miRNAs with significant reported associations with patient survival were found (Appendix A).

Considering the miRNAs with concordant significant results in more than two studies, miR-222-3p, and miR-155-5p were identified. Up-regulation of miR-222-3p [45,48,49] and miR-155-5p [30,38,46] was associated with a worse outcome in three different studies in each case. However, four and eight studies, respectively, did not find any association with prognosis for miR-222-3p [27,32,41,47] and miR-155-5p [26,27,32,33,41,47,48,49] (Table 3).

### 2.5. Pathway Enrichment Analysis

Predicted target genes for the miRNA that presented the highest evidence of being involved in DLBCL diagnosis (miR-155-5p and miR-21-5p) and in subtype classification (miR-155-5p, miR-221-3p) were identified in silico. Using these lists of genes, we searched for over-represented pathways that could be linked to DLBCL.

MAPK signaling pathway—Homo sapiens (KEGG) was significantly enriched among the predicted target genes of the two miRNAs associated with DLBCL diagnosis (Appendix A). Regarding the targets of the two miRNAs associated with DLBCL subtype, signaling by receptors tyrosine kinases (Reactome) was significantly enriched (Appendix A). The predicted target genes covered more than 20% of the genes included in these two pathways (Table 4).

## 3. Discussion

In this systematic review, we have performed an in depth analysis of the current literature in relation to the potential role of miRNA expression in tumor biopsies as biomarker for diagnosis, subtype characterization, treatment response and prognosis in patients with DLBCL.

Regarding the suitability of miRNAs as diagnostic biomarkers in DLBCL, twenty one articles were identified, in which a total of four miRNAs (miR-155-5p [17,22,24,26,28,30,32,33,34], miR-21-5p [19,22,26,28,33,36], miR-150-5p [18,28,29,31,32]) and miR-146a/b-5p [21,22,27,30] were found to be significantly deregulated in DLBCL patients in more than two studies with concordant results. Among them, miR-155-5p and miR-21-5p presented the most consistent results, being found upregulated in DLBCL patients in most studies.

MiR-155-5p, was the most widely studied miRNA and was found to be upregulated in DLBCL patients in nine of the studies in which it was analyzed [17,22,24,26,28,30,32,33,34] while no significant association was found in the other two studies [27,29]. Among the two studies that did not find a significant association between mir-155-5p and DLBCL, one presented the smallest sample size with nineteen patients [29], and the other study followed stricter criteria for statistically significant associations [27]. In agreement with these results, previous studies have suggested that miR-155 could represent an onco-miR as its expression is activated in many tumors, i.e., prostate cancer, breast cancer, and other tumors, particularly those of the lymphoid tissue [50,51,52]. A possible explanation for its implication in DLBCL is that the validated targets of this miRNA include known hallmarks of DLBCL, such as *SOSC* or *SHIP1* [53].

On the other hand, it is noteworthy that miR-21-5p, which was analyzed in eight independent studies, was significantly upregulated in DLBCL patients in six of them [19,22,26,28,33,36], while no statistically significant association was found in the other two studies [27,32]. In agreement with this observation, miR-21 has been reported to be deregulated in most cancers, such as colorectal cancer, acting as an oncogene [54]. High levels of miR-21 have also been observed in B-NHLs. Overall, miR-21 is considered to be an onco-miR that acts through the inhibition of the expression of different phosphatases, such as PDCD4 (Programmed Cell Death 4) and PTEN (Phosphatase And Tensin Homolog), which control the activity of signaling pathways like AKT and MAPK [55].

Given that miR-155-5p and miR-21-5p seem the best candidates as putative diagnostic tools in patients with DLBCL, their functional implication was inferred by in silico analysis. This analysis showed that MAPK signaling pathway is over-represented among the combined predicted target genes of miR-155-5p and miR-21-5p (Table 4). Interestingly, the genes predicted to be targeted by miR-155-5p and miR-21-5p are in the first steps of the signaling cascade (*CACN, RTK, IL1R* or *NIK*). Aberrant expression of this pathway is a major and highly prevalent oncogenic event in many human cancers [56], including NHL [57], which could explain the role of these miRNAs in DLBCL. In this regard, miR-21-5p is also one of the most frequently upregulated circulating microRNAs previously described as a non-invasive diagnosis biomarker [9].

The utility of microRNAs for DLBCL classification has been analyzed by twenty studies. A total of five miRNAs (miR-155-5p [26,27,28,30,33,34,38,41,44], miR-221-3p [27,33,41,45], miR-222-3p [27,41,45], miR-146a/b-5p [28,41,43], and miR-28-5p [27,38,43,45]) were found to be deregulated in more than two studies. However, miR-222-3p, miR-146a/b-5p, and miR-28-5p showed contradictory results since they were not found to be significantly related to DLBCL classification in four [28,32,38,43], five [21,27,28,38,43] and two studies [28,41], respectively. Some of the discrepancies might be due to the fact that subtype classification of the DLBCL patients was performed by GEP or IHC, which makes the studies less comparable due to the variable reproducibility of IHC stains and interpretations. The only miRNAs that showed more consistent results were miR-155-5p and miR-221-3p. MiR-155-5p was found to be upregulated in the ABC subgroup in nine out of ten studies and only found to be not associated in a study which used IHC for classification and a more stringent requirement for differentially expressed miRNAs [32]. Mir-221-3p was found to be upregulated in the ABC subgroup in four of the six studies in which it was analyzed. 

Taking into account the two most consistent microRNAs related with DLBCL classification, miR-155-5p and miR-221-3p, in silico analysis showed that the Tyrosine Kinase pathway was over-represented among their predicted target genes (Table 4). Among the target genes of both microRNAs, we found *PIK3R1* (p85), which is a negative regulator of the phosphatidylinositol 3-kinase (PI3K)-AKT pathway. Our data could indicate that overexpression of miR-155-5p and miR-221-3p in ABC subgroup repressed *PIK3R1* (p85), the PIK regulatory subunit, activating the PI3K-AKT signaling pathway in this subtype. However, it should be noted that it would be difficult to classify different DLBCL subtypes simply based on those two miRNAs. Thus, additional molecular biomarkers would be needed for clinical application.

Focusing on miRNAs as predictive biomarkers of response to R-CHOP treatment, five studies were identified with no agreement in the miRNAs considered [18,23,24,30,42]. Among them, upregulation of miR-27-3p [18], miR-34a-5p [42] and miR-224-5p [23] were associated with chemosensitivity and miR-155-5p and miR-146-5p [30] were associated with chemoresistance (Appendix A). Further studies are needed to confirm these preliminary results.

Finally, the implications of microRNAs in prognosis in DLBCL has been analyzed in nineteen studies including 50 significant miRNAs [18,20,21,23,25,26,27,30,32,33,41,42,45,46,47,48,49]. Among them, the expression of miR-222-3p [45,48,49], and miR-155-5p [30,38,46] were found to be associated with prognosis in more than two studies with concordant results. However, these miRNAs were analyzed in an equal or higher number of additional studies without finding any association with prognosis, which means that none of the analyzed miRNAs were established as a reliable marker of prognosis. It is noteworthy that most studies failed to report the specific treatment regimens, which would be of relevance in order to find prognostic biomarkers since prognosis is dependent on the specific treatment regimen. 

Several limitations were faced while performing this systematic review. On the one hand, the studies performed usually considered a limited set of selected miRNAs, which limits the number of comparable results and centers the discussion on those miRNAs that are better known, leaving other miRNAs aside. It is necessary to perform large-scale studies with a wider array of miRNAs using techniques such as next-generation sequencing that allow the identification of new miRNAs. On the other hand, most studies analyzed in this revision relied on tissue-based miRNA detection using qRT-PCR. As a result, it is difficult to know whether the differentially expressed miRNAs directly result from DLBCL or from the cancer-associated microenvironment. Single-cell RNA sequencing methods, developed in recent years, may provide a better approach to achieve this goal in future studies. Further, the included studies present great heterogeneity in sample sources, types of controls used or methodology for expression analysis. This methodological variability could be a source of differences in results among studies. Since the effect of such differences is difficult to determine in the context of a review, it would be of great relevance to reach a consensus and standardize the methodology of study used for future studies in order to facilitate reproducibility and comparisons among studies.

In addition, there is variability in the cut-off value for statistical significance among studies, which we considered to be a potential source of heterogeneity. Finally, there is a tendency to only publish statistically significant results, which leads to bias. All of these limitations in the published literature may be contributing to the lack of consistency in many of the results, which makes it difficult to draw final conclusions about the role of some of the miRNAs analyzed as biomarkers in DLBCL.

## 4. Materials and Methods

### 4.1. Systematic Review

#### 4.1.1. Search Strategy

A systematic search with the terms “((‘Non-coding RNA’) OR (‘miRNA’ OR ‘microRNA’ OR ‘miR’) OR (‘exosome’) OR (‘extracellular vesicle’) OR (‘secretome’)) AND (‘Diffuse large B cell lymphoma’ OR ‘DLBCL’)”, following the same strategy used in our previously published review on circulating miRNAs [9] was performed using the PubMed database (https://www.ncbi.nlm.nih.gov/pubmed/), including articles published until December 2017.

#### 4.1.2. Inclusion and exclusion criteria

Independent original studies that evaluated the expression of miRNAs in DLBCL tumor tissue as diagnosis, subtype, prediction of treatment response or prognosis biomarkers in human patient populations were included. Exclusion criteria encompassed: articles not including original data (reviews, meta-analyses, letters, and comments), case reports, abstracts, articles not published in English, studies that did not include miRNA data on human populations, and studies on diseases other than DLBCL. After full text revision, articles that included other diseases, analyzed circulating miRNAs, were focused on non-primary DLBCL, did not assess the role of miRNAs in diagnosis, subtype, treatment response, or prognosis, or did not analyze miRNA expression, were excluded. References within the identified studies were reviewed to identify additional matches. Study selection was performed by two researchers independently (AL and BS) and disagreements were resolved by consensus.

#### 4.1.3. Data Extraction

The following information was extracted from each study: publication year, type of tissue sample analyzed, characteristics of the study population, methodology, number of miRNAs studied, and the list of differentially expressed miRNAs provided. Only the miRNAs that were reported as statistically significant in more than two studies with consistent results were selected. 

### 4.2. Data Analysis

#### 4.2.1. Target Genes Selection

In order to predict the putative target genes for the miRNAs identified in the systematic search, miRWalk 2.0 database (http://zmf.umm.uni-heidelberg.de/apps/zmf/mirwalk2/) [58] was used. Only those genes predicted by 6 or more of the 12 available prediction algorithms available at miRWalk 2.0 were taken into account.

#### 4.2.2. Pathway Enrichment Analysis

In order to analyze pathways enrichment within the lists of predicted target genes, the over-representation analysis module of the ConsensusPathDB web tool (CPdB) (http://consensuspathdb.org/) was used [59]. Within this tool, KEGG (https://www.genome.jp/kegg/) [60], Reactome (https://reactome.org/PathwayBrowser/) [61] and BioCarta (http://cgap.nci.nih.gov/Pathways/BioCarta_Pathways) databases were interrogated, assuming a conservative *p*-value cutoff of 0.0001.

## 5. Conclusions

In this systematic review, we have identified that the expression of miR-155-5p and mir-21-5p shows the potential for utility in diagnosis, while mir-155-5p and mir-21-5p could be of use for DLBCL classification. Nevertheless, other associations between miRNA expression and DLBCL phenotypes showed contradictory results. We can conclude that this is a very promising field of study, which could also help to identify novel therapeutic targets and strategies to guide treatment choice. In order to exploit the potential of this field, it would be of particular interest to perform large-scale studies with large sample sizes and a wider array of miRNAs, including unknown miRNAs.

## Figures and Tables

**Figure 1 cancers-11-00144-f001:**
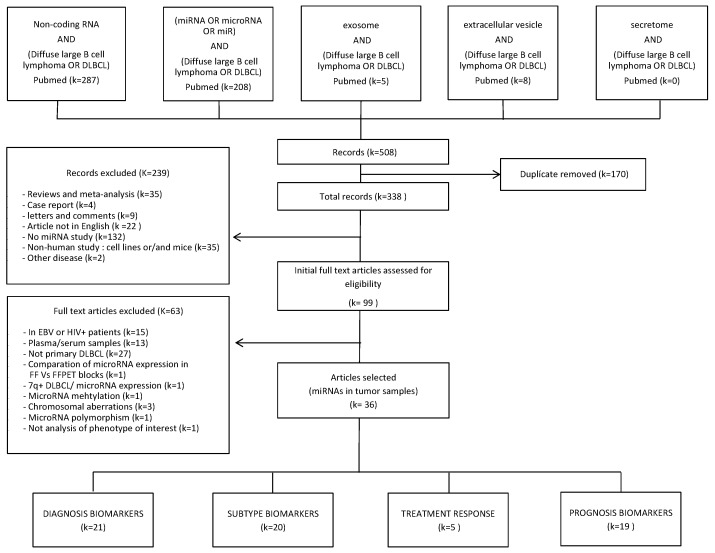
Flow-chart diagram of study selection. k = number of records.

**Table 1 cancers-11-00144-t001:** miRNAs significantly associated with DLBCL diagnosis in more than two studies.

Significant miRNAs	Result	*n* DLBCL	*n* Control	Sample Source	Method	n miRNAs	Reference
**miR-155-5p**	up	29	32 (RLH)	Tissue	qRT-PCR	1	Li et al. 2017 [17]
		22	6 (NLN)	Biopsie	qRT-PCR	1	Huskova et al. 2015 [24]
		200	11 (NT)	FFPE	qRT-PCR	3	Go et al. 2015 [26]
		45 (DC);75 (VC)	10 (DC);6 (VC) (NLN)	FF and FFPE	qRT-PCR/array	177	Caramuta et al. 2013 [28]
		90	31 (RLN)	FFPE	qRT-PCR	2	Zhong et al. 2012 [30]
		58	7 (NLN)	FFPE	qRT-PCR	157	Roehle et al. 2008 [32]
		48	6 (NBC)	FF and FFPE	qRT-PCR	3	Lawrie et al. 2007 [33]
		23	2	FF	Semi RT-PCR	1	Eis et al. 2005 [34]
		24	14 (NLN)	FFPE	array	3100 probes	Tamaddon et al. 2016 [22]
	NS	92	15	FF	sequencing	miRNAome	Lim 2015 et al. [27]
		12	7	FFPE	qRT-PCR	4	Handal et al. 2013 [29]
**miR-21-5p**	up	55	20 (NLN)	FF and FFPE	qRT-PCR	1	Liu et al. 2017 [19]
		26	10 (NLN)	FFPE	qRT-PCR	1	Song et al. 2017 [36]
		200	11 (NT)	FFPE	qRT-PCR	3	Go et al. 2015 [26]
		45 (DC);75 (VC)	10 (DC);6 (VC)(NLN)	FF and FFPE	qRT-PCR/array	177	Caramuta et al. 2013 [28]
		48	6 (NBC)	FF and FFPE	qRT-PCR	3	Lawrie et al. 2007 [33]
		24	14 (NLN)	FFPE	array	3100 probes	Tamaddon et al. 2016 [22]
	NS	92	15	FF	sequencing	miRNAome	Lim et al. 2015 [27]
		58	7 (NLN)	FFPE	qRT-PCR	157	Roehle et al. 2008 [32]
**miR-150-5p**	up	12	7	FFPE	qRT-PCR	4	Handal et al. 2013 [29]
	down	45 (DC);75 (VC)	10 (DC);6 (VC)(NLN)	FF and FFPE	qRT-PCR/array	177	Caramuta et al. 2013 [28]
		36	5 (NLN)	Tissue	qRT-PCR	8	Fassina et al. 2012 [31]
		58	7 (NLN)	FFPE	qRT-PCR	157	Roehle et al. 2008 [32]
		5	4 (RLH)	Tissue	nanostring	800	Jia et al. 2017 [18]
	NS	92	15	FF	sequencing	miRNAome	Lim et al. 2015 [27]
**miR-146b-5p**	down	106	30 (RLH)	FFPE	qRT-PCR	939	Wu et al. 2014 [21]
	NS	45 (DC);75 (VC)	10 (DC);6 (VC)(NLN)	FF and FFPE	qRT-PCR/array	177	Caramuta et al. 2013 [28]
	up	92	15	FF	sequencing	miRNAome	Lim et al. 2015 [27]
		24	14 (NLN)	FFPE	array	3100 probes	Tamaddon et al. 2016 [22]
**miR-146a-5p**		90	31 (RLN)	FFPE	qRT-PCR	2	Zhong et al. 2012 [30]
		24	14 (NLN)	FFPE	array	3100 probes	Tamaddon et al. 2016 [22]
	NS	45 (DC);75 (VC)	10 (DC);6 (VC)(NLN)	FF and FFPE	qRT-PCR/array	177	Caramuta et al. 2013 [28]
		92	15	FF	sequencing	miRNAome	Lim et al. 2015 [27]

RLH: Reactive lymphoid hyperplasia; NLN: normal lymph node tissues; NT: normal tonsil; FF: fresh frozen; FFPE: formalin-fixed paraffin-embedded; DC: discovery cohort; VC: validation cohort; NBC: normal B cell samples; Up: statistically significantly upregulated in DLBCL patients; NS: no significant difference between patients and controls; Down: significantly downregulated in DLBCL patients.

**Table 2 cancers-11-00144-t002:** miRNAs significantly associated with DLBCL subtype in more than two studies.

Significant miRNAs	Result	*n* GCB	*n* ABC	Sample Source	Method	n miRNAs	Reference
**miR-155-5p**	Down GCB	53	95	FFPE	qRT-PCR	8	Go et al. 2015 [26]
		32	27	FFPE	qRT-PCR/array	377	Iqbal et al. 2015 [38]
		20	34	FF and FFPE	qRT-PCR/array	177	Caramuta et al. 2013 [28]
		36	31	FF	qRT-PCR	1	Huang et al. 2012 [44]
		21	69	FFPE	qRT-PCR	2	Zhong et al. 2012 [30]
		32	28	FFPE	Array	464	Lawrie et al. 2009 [41]
		16	18	FF and FFPE	qRT-PCR	3	Lawrie et al. 2007 [33]
		4	19	FF	Semiq. RT-PCR	1	Eis et al. 2005 [34]
		41	30	FF	sequencing	miRNAome	Lim et al. 2015 [27]
	NA	25	25	FFPE	qRT-PCR	157	Roehle et al. 2008 [32]
**miR-221-3p**	Down GCB	11	18	FFPE	qRT-PCR/array	470	Montes-Moreno et al. 2011 [45]
		32	28	FFPE	Array	464	Lawrie et al. 2009 [41]
		16	18	FF and FFPE	qRT-PCR	3	Lawrie et al. 2007 [33]
		41	30	FF	sequencing	miRNAome	Lim et al. 2015 [27]
	NS	20	20	Tissue	Array	113	Zhang et al. 2009 [43]
		20	34	FF and FFPE	qRT-PCR/array	177	Caramuta et al. 2013 [28]
**miR-222-3p**	Down GCB	11	18	FFPE	qRT-PCR/array	470	Montes-Moreno et al. 2011 [45]
		32	28	FFPE	Array	464	Lawrie et al. 2009 [41]
		41	30	FF	sequencing	miRNAome	Lim et al. 2015 [27]
	NS	25	25	FFPE	qRT-PCR	157	Roehle et al. 2008 [32]
		20	20	Tissue	Array	113	Zhang et al. 2009 [43]
		32	27	FFPE	qRT-PCR/array	377	Iqbal et al. 2015 [38]
		20	34	FF and FFPE	qRT-PCR/array	177	Caramuta et al. 2013 [28]
**miR-146a-5p**	Down GCB	20	34	FF and FFPE	qRT-PCR/array	177	Caramuta et al. 2013 [28]
		21	69	FFPE	qRT-PCR	2	Zhong et al. 2012 [30]
	NS	41	30	FF	sequencing	miRNAome	Lim et al. 2015 [27]
		20	20	Tissue	Array	113	Zhang et al. 2009 [43]
**miR-146b-5p**	Down GCB	32	28	FFPE	Array	464	Lawrie et al. 2009 [41]
	NS	32	27	FFPE	qRT-PCR/array	377	Iqbal et al. 2015 [38]
		41	30	FF	sequencing	miRNAome	Lim et al. 2015 [27]
		20	20	Tissue	Array	113	Zhang et al. 2009 [43]
		47	59	FFPE	qRT-PCR	2	Wu et al. 2014 [21]
		20	34	FF and FFPE	qRT-PCR/array	177	Caramuta et al. 2013 [28]
**miR-28-5p**	Up GCB	11	18	FFPE	qRT-PCR/array	470	Montes-Moreno et al. 2011 [45]
		32	27	FFPE	qRT-PCR/array	377	Iqbal et al. 2015 [38]
		41	30	FF	sequencing	miRNAome	Lim et al. 2015 [27]
		20	20	Tissue	Array	113	Zhang et al. 2009 [43]
	NS	20	34	FF and FFPE	qRT-PCR/array	177	Caramuta et al. 2013 [28]
		32	28	FFPE	Array	464	Lawrie et al. 2009 [41]

GCB: Germinal center B-cell like; FF: fresh frozen; FFPE: formalin-fixed paraffin-embedded; NA: not available; Up: statistically significantly upregulated in the subtype of DLBCL patients; NS: no significant difference between patient subtypes; Down: significantly downregulated in the subtype of DLBCL patients.

**Table 3 cancers-11-00144-t003:** miRNAs significantly associated with DLBCL prognosis in more than two studies.

Significant miRNAs	Result	*n* DLBCL	Sample Source	Method	n miRNAs	Reference
**miR-222-3p**	Up: ↓OS	176	FFPE	qRT-PCR	11	Alencar et al. 2011 [48]
	Up: ↓PFS and OS	36/240	FFPE	qRT-PCR/array	470/9	Montes-Moreno et al. 2011 [45]
	Up: ↓OS and PFS	106	FFPE	qRT-PCR	3	Malumbres et al. 2009 [49]
	NS	64	FFPE	Array	464	Lawrie et al. 2009 [41]
		92	FF	sequencing	miRNAome	Lim et al. 2015 [27]
		58	Biopsie	qRT-PCR	157	Roehle et al. 2008 [32]
		83	FFPE	qRT-PCR/array	±900	Shepshelovich et al. 2015 [47]
**miR-155-5p**	Up: ↓survival	118	FF	qRT-PCR	1	Zhu et al. 2016 [46]
	Up: ↓OS	79	FFPE	qRT-PCR	8	Iqbal et al. 2015 [38]
	Down: ↑PFS	90	FFPE	qRT-PCR	2	Zhong et al. 2012 [30]
	NS	176	FFPE	qRT-PCR	11	Alencar et al. 2017 [48]
		200	FFPE	qRT-PCR	3	Go et al. 2015 [26]
		35	FF and FFPE	qRT-PCR	3	Lawrie et al. 2007 [33]
		64	FFPE	Array	464	Lawrie et al. 2009 [41]
		92	FF	sequencing	miRNAome	Lim et al. 2015 [27]
		106	FFPE	qRT-PCR	3	Malumbres et al. 2009 [49]
		58	Biopsie	qRT-PCR	157	Roehle et al. 2008 [32]
		83	FFPE	qRT-PCR/array	±900	Shepshelovich et al. 2015 [47]

FF: fresh frozen; FFPE: formalin-fixed paraffin-embedded; OS: overall survival; PFS: progression-free survival; EFS: event free survival; RFS: relapse free survival; Up: statistically significantly upregulated expression; NS: no significant association between expression and patient outcome; Down: significantly downregulated expression. ↓: decreased; ↑: increased.

**Table 4 cancers-11-00144-t004:** Coverage of DLBCL related pathway with miRNA predicted targets.

Phenotype	Pathway Name	miRNA	*p*-value	FDR	n Target Genes	Coverage
	Num. of Genes					
	Database					
**DLBCL Diagnosis**	MAPK signaling pathway (Homo sapiens) 295 genes (KEGG)	miR-155-5p miR-21-5p	3.42 × 10^−07^	0.000293	73	24.7%
**DLBCL** **Subtype**	Signaling by Receptor Tyrosine Kinases 422 genes (Reactome)	miR-155-5p miR-221-3p	1.09 ×10^−07^	6.25 × 10^−05^	103	24.4%

*p*-value: absolute *p*-value; FDR: corrected *p*-value by “False Discovery Rate” method; pathway source: associated database.

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
