# Peer review of "Systematic Review of the Potential of MicroRNAs in Diffuse Large B Cell Lymphoma"

_cancers, 2019, doi:10.3390/cancers11020144_

Round 1
Reviewer 1 Report
Diffuse large B cell lymphoma (DLBCL) is the most common type of non Hodgkin's lymphoma (NHL). Although it could be cured, more than half of the patients were resistant to chemotherapy and the diseases presented heterogeneous without validated biomarkers. In the recent studies, microRNAs have been suggested as useful biomarkers in cancer. The authors in this manuscript used systematic text mining approaches to look into the role of miRNAs in the diagnosis, classification, treatment response and prognosis of DLBCL patients. After classification, they concluded that miR-155-5p and mir-21-5p are two highly expressed miRNAs and could be used for DLBCL diagnosis, while miR-155-5p and miR-221-3p may be good markers for classification of different DLBCL types.
We appreciated that the authors comprehensively collected current studies about DLBCL-associated miRNAs and systematically evaluated them as useful biomarkers for DLBCL. Their suggestion and conclusion are quite significant and await further validation. Some concerns about their manuscript and collected studies are addressed as follows.
Major comments:
1. Most studies in this manuscript relied on the tissue-based miRNA detection using qPCR. It is difficult to know whether the differentially expressed miRNAs directly results from DLBCL or the cancer-associated microenvironment. Although this is an inevitable technical issue in the current studies and in this field, the authors should discuss this in their manuscript. For example, the single-cell RNA sequencing may be a better approach to achieve this goal and has been developed in the past few years.
2. It would be difficult to classify different cancer sub-types simply based on one or two genes or miRNAs. Although the authors summarized some distinct miRNAs preferentially expressed in either germinal center B-cell like DLBCL or the activated B-cell-like (ABC) DLBCL, a defined gene or miRNA set will be an appropriate indicator for clinical application.
Minor comments:
1. The miRNAs summarized in this manuscript are not novel in the Diffuse large B cell lymphoma biomarker field. Therefore, the authors may want to rephrase their wording in the introduction. For examples, they should claim that those miRNAs presented in this manuscript are more systematically-defined miRNAs for DLBCL classification.
2. The abbreviation of germinal center B-cell like DLBCL (GBC or GCB?) should be consistent in the table 2 and main text.
Author Response
Comments and Suggestions for Authors
Diffuse large B cell lymphoma (DLBCL) is the most common type of non Hodgkin's lymphoma (NHL). Although it could be cured, more than half of the patients were resistant to chemotherapy and the diseases presented heterogeneous without validated biomarkers. In the recent studies, microRNAs have been suggested as useful biomarkers in cancer. The authors in this manuscript used systematic text mining approaches to look into the role of miRNAs in the diagnosis, classification, treatment response and prognosis of DLBCL patients. After classification, they concluded that miR-155-5p and mir-21-5p are two highly expressed miRNAs and could be used for DLBCL diagnosis, while miR-155-5p and miR-221-3p may be good markers for classification of different DLBCL types.
We appreciated that the authors comprehensively collected current studies about DLBCL-associated miRNAs and systematically evaluated them as useful biomarkers for DLBCL. Their suggestion and conclusion are quite significant and await further validation. Some concerns about their manuscript and collected studies are addressed as follows.
Major comments:
1. Most studies in this manuscript relied on the tissue-based miRNA detection using qPCR. It is difficult to know whether the differentially expressed miRNAs directly results from DLBCL or the cancer-associated microenvironment. Although this is an inevitable technical issue in the current studies and in this field, the authors should discuss this in their manuscript. For example, the single-cell RNA sequencing may be a better approach to achieve this goal and has been developed in the past few years.
Answer: The technological limitation raised by the reviewer is of great interest in the field. In agreement with this comment, we have added the following information to the 10th paragraph of the Discussion:
“On the other hand, most studies analyzed in this revision relied on tissue-based miRNA detection using qRT-PCR. As a result, this makes difficult to know whether the differentially expressed miRNAs directly result from DLBCL or from the cancer-associated microenvironment. Single-cell RNA sequencing methods, developed in recent years, may provide a better approach to achieve this goal in future studies.”
2. It would be difficult to classify different cancer sub-types simply based on one or two genes or miRNAs. Although the authors summarized some distinct miRNAs preferentially expressed in either germinal center B-cell like DLBCL or the activated B-cell-like (ABC) DLBCL, a defined gene or miRNA set will be an appropriate indicator for clinical application.
Answer: We kindly appreciate the reviewer’s comment. In order to improve the discussion, following the reviewer´s recommendation, we have included the following sentence in the 7th paragraph of the Discussion:
“However, it should be noted that it would be difficult to classify different DLBCL subtypes simply based on those two miRNAs. Thus, additional molecular biomarkers would be needed for clinical application.”
Minor comments:
1. The miRNAs summarized in this manuscript are not novel in the Diffuse large B cell lymphoma biomarker field. Therefore, the authors may want to rephrase their wording in the introduction. For examples, they should claim that those miRNAs presented in this manuscript are more systematically-defined miRNAs for DLBCL classification.
Answer: Following the reviewer´s suggestion, we have reworded the last sentence of the Introduction:
“Consequently, the aim of this systematic review was to clarify the role of deregulated miRNAs in DLBCL tumor samples as more systematically-defined diagnostic, subtype, prediction of treatment response and prognostic biomarkers.”
2. The abbreviation of germinal center B-cell like DLBCL (GBC or GCB?) should be consistent in the table 2 and main text.
Answer: We greatly appreciate this comment and we have now standardized the abbreviation of germinal center B-cell like DLBCL as “GCB”.
Reviewer 2 Report
In this systematic review, the authors aimed to porvide new insight into the role of miRNAs in the diagnosis, classification,treatment response and prognosis of DLBCL patients.
In this paper, the authors wrote the results of all the studies they found, but without any analyses. They do not discuss about the different methods used, the different controls (RLH, NLN...), FFPE or FF.... Table are usually not clear. The comparisons of miRNA expression in DLBCL and RLH must be discussed.
This manuscript lacks analysis/discussion of the results of the previous published articles.
To my opinion, results part and discussion part should not be separated: there are many repetions in the discussion part.
Author Response
Comments and Suggestions for Authors
In this systematic review, the authors aimed to porvide new insight into the role of miRNAs in the diagnosis, classification,treatment response and prognosis of DLBCL patients.
In this paper, the authors wrote the results of all the studies they found, but without any analyses. They do not discuss about the different methods used, the different controls (RLH, NLN...), FFPE or FF.... Table are usually not clear. The comparisons of miRNA expression in DLBCL and RLH must be discussed.
This manuscript lacks analysis/discussion of the results of the previous published articles.
Answer: In order to address the reviewer´s comments we are going to focus on two different aspects.
On the one hand, regarding the lack of analyses addressed by the reviewer, we can argue that, due to the heterogeneity on the type of results presented in each of the reviewed articles, we considered that carrying out any kind of meta-analysis would be very difficult to carry out and the results would be biased. As a result, we decided on performing a systematic review (without meta-analyses).
On the other hand, focusing on the lack of discussion on the different methods used, controls, or sample sources, as brought up by the reviewer, we would like to say that we did not see a clear effect of any on this variability on the differences in results among studies. In fact, we do see a possible effect of sample size, method for subtype determination or data analysis methodology and that is the reason why those factors are further considered in the discussion, instead of those mentioned above. Furthermore, it is difficult to establish, based on the available literature and within the scope of this review, which would be the best method, sample source… to use, since interesting results are obtained with all of them. Yet, it is true that those factors that the reviewer raises are a source of variability that should be taken into account anyway and one of the limitations of this study. Accordingly, we have added the following information to the 10th paragraph of the Discussion:
“Further, the included studies present a great heterogeneity in sample sources, types of controls used or methodology for expression analysis. This methodological variability could be a source of differences in results among studies. Since the effect of such differences is difficult to determine in the context of a review, it would be of great relevance to reach a consensus and standardize the methodology of study used for future studies in order to facilitate reproducibility and comparisons among studies.”
To my opinion, results part and discussion part should not be separated: there are many repetions in the discussion part.
Answer: We appreciate the reviewer’s suggestions but, following the standard PRISMA guidelines for systematic reviews (Moher D, Liberati A, Tetzlaff J, Altman DG, The PRISMA Group (2009). Preferred Reporting Items for Systematic Reviews and Meta-Analyses: The PRISMA Statement. PLoS Med 6(7): e1000097. doi:10.1371/journal.pmed1000097), the most widely accepted format for systematic reviews includes separated sections for Results and Discussion.
Reviewer 3 Report
The authors performed a systematic review on th role of miRNAs in tumor samples of DLBCL. Data are presented clearly, the manuscript is well written,
The authors recently published a systematic review on circulating miRNs in DLBCL using the same literature search criteria (Lopez-Santillana et al, Oncotarget 2018, 9: 22850).
In the abstract the authors should clearly state that the current analysis is restricted to the role of miRNAs in tumor samples and does not consider circulating miRNAs, the topic of the previous review. A literature review including the terms exososomes otherwise raises the expectation to find also a review on circulating miRNAs. The authors should reference their recent review.
Minor comments:
Adding the reference number to the name of the authors in the tables could be helpful to identify the publications.
line 170: "two studies" instead of "three studies"
line 231: "known" in stead of "know"
Author Response
Comments and Suggestions for Authors
The authors performed a systematic review on th role of miRNAs in tumor samples of DLBCL. Data are presented clearly, the manuscript is well written,
The authors recently published a systematic review on circulating miRNs in DLBCL using the same literature search criteria (Lopez-Santillana et al, Oncotarget 2018, 9: 22850).
In the abstract the authors should clearly state that the current analysis is restricted to the role of miRNAs in tumor samples and does not consider circulating miRNAs, the topic of the previous review. A literature review including the terms exososomes otherwise raises the expectation to find also a review on circulating miRNAs. The authors should reference their recent review.
Answer: We appreciate the reviewer’s suggestions and have included the suggested information in the abstract:
“As a result, the analysis was restricted to the role of miRNAs in tumor tissue and we did not consider circulating miRNAs”.
We have also included a reference to our previous review in the Methods: Search strategy:
“A systematic search with the terms [((“Non-coding RNA”) OR (“miRNA” OR “microRNA” OR “miR”) OR (“exosome”) OR (“extracellular vesicle”) OR (“secretome”)) AND (“Diffuse large B cell lymphoma” OR “DLBCL”)], following the same strategy used in our previously published review on circulating miRNAs (9), was performed using PubMed database (https://www.ncbi.nlm.nih.gov/pubmed/), including articles published until December 2017.
Minor comments:
Adding the reference number to the name of the authors in the tables could be helpful to identify the publications.
Answer: Following the reviewer´s suggestion, we have included the reference numbers in each of the tables.
line 170: "two studies" instead of "three studies"
Answer: We thank the reviewer for the attention to these details and we have modified this mistake in the Discussion.
line 231: "known" in stead of "know"
Answer: Following the reviewer´s indications, we have modified this grammatical error in the Discussion.
Round 2
Reviewer 2 Report
The authors globally answer all requests